# Liposome-Based Drug Delivery Systems in Cancer Immunotherapy

**DOI:** 10.3390/pharmaceutics12111054

**Published:** 2020-11-04

**Authors:** Zili Gu, Candido G. Da Silva, Koen Van der Maaden, Ferry Ossendorp, Luis J. Cruz

**Affiliations:** 1Department of Radiology, Leiden University Medical Center, Albinusdreef 2, 2333 ZA Leiden, The Netherlands; z.gu@lumc.nl (Z.G.); C.da_Silva@lumc.nl (C.G.D.S.); 2Tumor Immunology Group, Department of Immunology, Leiden University Medical Center, Albinusdreef 2, 2333 ZA Leiden, The Netherlands; K.van_der_Maaden@lumc.nl (K.v.d.M.); F.A.Ossendorp@lumc.nl (F.O.); 3TECOdevelopment GmbH, 53359 Rheinbach, Germany

**Keywords:** liposome, drug delivery, cancer immunotherapy, immunomodulation

## Abstract

Cancer immunotherapy has shown remarkable progress in recent years. Nanocarriers, such as liposomes, have favorable advantages with the potential to further improve cancer immunotherapy and even stronger immune responses by improving cell type-specific delivery and enhancing drug efficacy. Liposomes can offer solutions to common problems faced by several cancer immunotherapies, including the following: (1) Vaccination: Liposomes can improve the delivery of antigens and other stimulatory molecules to antigen-presenting cells or T cells; (2) Tumor normalization: Liposomes can deliver drugs selectively to the tumor microenvironment to overcome the immune-suppressive state; (3) Rewiring of tumor signaling: Liposomes can be used for the delivery of specific drugs to specific cell types to correct or modulate pathways to facilitate better anti-tumor immune responses; (4) Combinational therapy: Liposomes are ideal vehicles for the simultaneous delivery of drugs to be combined with other therapies, including chemotherapy, radiotherapy, and phototherapy. In this review, different liposomal systems specifically developed for immunomodulation in cancer are summarized and discussed.

## 1. The Potential of Immunotherapy for the Treatment of Cancer

Cancer immunotherapy has been widely explored because of its durable and robust effects [1]. Tumors are more than just insular masses consisting of proliferating cancer cells; they have a complex composition built by multiple cell types, which participate in heterotypic interactions with each other [2]. Sustained antitumor responses triggered by immunotherapeutic treatments have been demonstrated via the active stimulation of specific targets such as immune cells, normalization of the tumor microenvironment (TME), and other mechanisms.

### 1.1. The Generation and Regulation of Tumor Immunity

The generation of clinically effective antitumor responses normally requires the successful execution of several immune processes (Figure 1). Firstly, numerous cancer antigens, either tumor-specific or tumor-associated antigens (TAAs), are released during the process of tumor growth. These cancer antigens are phagocytosed, processed, and presented by antigen-presenting cells (APCs) such as dendritic cells (DCs). Then, the cancer antigens can be presented into the major histocompatibility complex (MHC) class II molecules or cross-presented into the MHC class I molecules on DCs that migrate to draining lymph nodes to initiate T cell activation [3]. During this process, DCs mature and costimulatory molecules (such as CD40, CD80, and CD86) are upregulated when specific cues are present, such as damage-associated molecular pattern molecules or pathogen-associated molecular pattern molecules present in the TME or provided by means of treatment. Upon maturation, DCs remodel their membranes to form dendrites to increase the membrane surface area and enhance T cell interactions [4]. Accordingly, higher numbers of DCs present in the TME are beneficial and can improve T cell activation [5,6].

Next, productive T cell responses are generated in lymphoid organs [7]. During this process, tolerance can still be promoted by regulatory T cells (Treg), and inhibitory receptors would oppose anti-tumor efficacy. As the potential site for therapeutic intervention, stimulatory adjuvants can be used to skew the magnitude and type of T cell response. Agonistic antibodies to secondary immune checkpoint molecules such as 4-1BB, OX40, and glucocorticoid-induced TNFR-related protein could amplify anti-tumor immune responses.

Once activated, effector T cells must migrate to the tumor site and infiltrate the TME to perform their killing job. Here, negative regulatory signals that dampen T cell activation or induce anergy and exhaustion must be avoided as much as possible [8]. Typically, cytotoxic T-lymphocyte-associated protein 4 (CTLA-4) and programmed cell death protein 1 (PD-1) expressed on activated T cells are major suppressive costimulatory molecules, and therapeutic disruption with antagonistic antibodies has shown strong therapeutic potential [9,10]. Inside the abnormal TME, tumor populations, stromal cells, and multitudes of innate and adaptive immune cells together build up a complicated network to help tumor escape immune attacks through a variety of mechanisms [11,12,13,14]. Hence, an interesting strategy is to augment the anti-tumor immune response to overcome diverse immunosuppressive signals, which may be driven by both suppressive mediators and regulatory cell populations [15,16]. In this review, we have summarized the therapeutic strategies of immunomodulation in recent years and discuss the different mechanisms used to intervene with tumor immunity through the application of liposome technology.

### 1.2. Recent Development of Cancer Immunotherapy

Immunotherapy refers to the approach to treat cancer through generating or regulating an immune response against it [1]. Recently, harnessing immunotherapy has been a fundamental strategy in cancer therapy. In last two decades, various types of immunotherapies were developed to improve anti-tumor response through the modulation of stimulatory, suppressive, or regulatory mechanisms. These strategies include vaccines, monoclonal antibodies, immunomodulatory small molecules, as well as the exploration of the immunomodulatory functions of chemotherapy and radiation therapy [17,18]. In order to generate a successful and powerful immune response, cancer vaccine is normally given to enhance the immune system. In accordance with the immune response steps against cancer, this approach focuses on (1) enhancing antigen uptake, processing and presentation to T cells, and hence enhancing the activation and expansion of naïve T cells, e.g., antigen/adjuvant vaccines or cytokines that promote APC functions; and (2) intensifying the effector phase of immune responses, such as infusing back ex vivo stimulated and expanded tumor infiltrate T cells to patients. It is noted that this strategy shows great supply to the immune activation process in patients, but it might also push the immune system to a supraphysiological level with an increased risk of immune-related adverse events [19]. For antibody-targeted therapy, various strategies including using antibodies to target cancer directly, altering the host response to cancer, and delivering cytotoxic substances to cancer have been investigated. Oncologists now see monoclonal antibody (mAb)-based cancer therapies as a vital component of state-of-the-art cancer care; one typical example is using mAbs to block B7-H1 and PD-1 interactions [20]. There are arrays of molecular pathways that cause immune defects in the TME that can be targeted to reset or reprogram anti-tumor immunity. Molecular entities and the mechanisms of these pathways could be potential targets for cancer immunotherapy and provide an alternative for patients that cannot benefit from current immunotherapy [21].

## 2. The Emergence of Liposomes as Drug Delivery Vehicles in Cancer Immunotherapy

The increasing research on the applicability of nanotechnology in cancer therapy is based on its unique hallmarks from the fields of drug delivery, diagnosis, and imaging [22]. Nanocarriers that incorporate these features are very promising for clinical applications, and a variety of them have been explored in (pre) clinical research stages. There are several different types of nanocarriers including liposomes, polymer micelles, inorganic nanoparticles, drug conjugates, and virus-like nanoparticles, which have been used for enhancing the drug delivery of chemotherapeutics, radiation therapy, gene therapy, and immunotherapy. Along with the enormous progress achieved in the field of immunotherapy, nanotechnology-based immunotherapy has gradually displayed potential to improve the limitation associated with therapeutic monoclonal antibodies, tyrosine kinase inhibitors, and cancer vaccines [23,24,25]. It has distinctive features such as improving drug efficacy, reducing toxicity, better physicochemical properties, the capacity to deliver macromolecular drugs, and the ability to bypass tumor-driven resistance mechanisms [26,27,28]. One such nanotechnology-based system for cancer immunotherapy is liposomes. Liposomal formulations have been used in many clinical trials [29] for different purposes (e.g., for cancer targeting and vaccination). One of these formulations is being used in the clinic for cancer treatment (i.e., Doxil^®^).

Liposomes are lipid-based nanoparticles with high potential to improve cancer immunotherapies, since they can incorporate and/or associate a high variety of cancer drug molecules (e.g., peptides, proteins, antibodies, low-molecular weight chemotherapeutics) [30,31]. Liposomes are very versatile because they can be used for different kinds of immunotherapeutic cancer treatments (e.g., vaccination and checkpoint blockade), as Figure 2 showed [32]. They are popular platforms for the controlled release of antigens, immunomodulators, and low-molecular-weight anti-cancer drugs [33]. The usage of liposomal-based drug delivery systems based in immunotherapy can be grouped into five different categories (Figure 3): (1) Vaccination: harnessing liposomes for the coordinated delivery of antigens and other stimulatory molecules to APCs or T cells, which employs the power of modern nanotechnology and yields improved outcomes as compared to conventional tumor antigen vaccination; (2) Tumor normalization: overcoming tumor-driven immunosuppressive signals (e.g., checkpoint blockade) in the TME by liposomes to improve selectivity and decrease systemic toxicity, which provides preliminary evidence of efficacy; (3) Tumor modulation: correcting or modulating an existing or known pathway during the development of the anti-tumor response; (4) Tumor targeting: targeting overexpressed surface molecules on cancer cells (may also be self-antigens) via B cell/antibody route or cancer-specific peptides presented on MHC-I on the cancer cells via Ag-specific T-cells, especially cytotoxic T-cells; and (5) Combinational therapy: exploring the combinational strategies between immunotherapy and others (e.g., chemotherapy, radiotherapy and phototherapy et al.), which provides opportunities for liposomes to co-load molecules with different properties.

Liposomes, in particular polyethylene glycol (PEG)ylated liposomes, tend to passively accumulate in tumor tissue via the enhanced permeability and retention (EPR) effect [34,35,36]. Conjugating a hydrophilic polymer to the surface of liposome reduces opsonization and clearance by the reticuloendothelial system [37]. The surfaces of liposomes are often modified with antibodies or specific receptor ligands to increase the binding of the liposomes to the target cells, which is regarded as a promising strategy for cancer treatment [38,39,40,41]. Liposome-mediated immunotherapy can be potentially used to mediate efficient delivery to target sites and provoke robust immune responses [42,43,44]. Since the tumor immunity plays such an important role in tumor development, progression, and metastasis, it offers opportunities for liposomes to improve the efficiency of cancer treatment. The continued development of liposomes is one of the essential aspects of the pursuit of safe and effective cancer immunotherapy.

In spite of improved biodistribution and tumor accumulation, there are also certain issues that need to be addressed to get the maximum benefit from current liposomal platforms. As the complexity of liposomes increases, so do the expenses and difficulties associated with their preparation and quality control. The physicochemical properties of liposomes, including their size, charge, polarity, and any modifications, may also have a negative impact on the ability of the liposomes to reach the tumor via the EPR effect [45,46]. For example, PEGylated liposomes bigger than 500 nm in diameter are rapidly removed from the blood by optimization. Although liposomes have emerged as a promising approach to overcome the limitations in current cancer treatment and have shown high efficiency in multiple animal models, they might not be sufficient when employed in cancer patients. Unlike in many murine cancer models, a considerable barrier resulting from imperfect or inefficient EPR effect contributes to limit the concentration, penetration, and distribution of liposomes in human tumors [47]. Liposomes should be designed and characterized on the basis of their interactions with complex transport barriers located in the TME [48]. In addition, it is also crucial to improve strategies to achieve strong antitumor effects while minimizing toxicity to normal cells. Herein, we assess the current status based on the current studies in the literature that have focused on liposome-mediated immunotherapy and immunomodulation, and we summarize the recently proposed strategies to overcome the limited immune responses and low efficacy-related issues in this field.

## 3. Liposome-Based Drug Delivery Systems Used for Immunotherapy

### 3.1. Liposome-Based Delivery of Stimulatory Molecules for Eliciting Immune Responses

It has been established that liposomes with immunostimulatory properties can induce potent immune responses [49]. Generally, the approaches that are employed to elicit or enhance immune responses through the modulation of regulatory mechanisms have been widely investigated in cancer immunotherapy for several decades [19]. These enhancement strategies are among the most powerful methods to achieve successful cancer immunotherapy. Recent studies have explored the ability of liposomes loaded with immune stimulatory molecules to augment the potency of cancer immunotherapy [50,51]. For example, the cancer treatment with “free” immunostimulatory anti-CD137 and interleukin (IL)-2 Fc fusion protein can enhance immune responses and improve anti-tumor activity, but at the same time, it can induce intolerable toxicity [52,53]. However, in multiple tumor models, the study of Zhang et al. showed that stealth (PEGylated) immunoliposomes, whose surface is conjugated with IL-12 and anti-CD137, had equivalent immunostimulatory effects compared to that of the “free” drugs, but with a nearly complete absence of systemic toxicity in a murine melanoma model [54]. These immunoliposomes showed rapid drug accumulation in tumor tissue and elicited potent immune activation, which allowed repetitive dosage of the immunoliposome forms for strong anti-tumor activity without systemic toxicity. Likewise, Kwong et al. demonstrated that effective anti-tumor immunity can be achieved with lower systemic toxicity after restricting the biodistribution of these immunotherapeutic agents by using liposomes [55]. Upon intratumoral injection, these liposomes tended to disseminate in the tumor parenchyma and tumor-draining lymph nodes instead of entering into systemic circulation. In animal experiments, liposomes showed less pathophysiological symptoms and elicited only minimal increases in systemic cytokines compared to the PBS control group, whereas anti-CD137 + IL 2 therapy induced significant elevations in the serum levels of inflammatory cytokines. These studies suggest that immunostimulatory therapies via liposomes can induce potent anti-tumor effects while eliminating systemic side effects, which can efficiently broaden the clinical application of immunostimulatory therapies by using liposome technology.

Hence, the efficient and targeted delivery of these stimulatory molecules to the cells of interest is one of the important keys for successful cancer immunotherapy. Compared to injecting “free” drugs directly, liposomes can prevent cargos from degrading in the surrounding biological environment, improve their biodistribution, and promote their delivery to target cells. There are two well-known targets to improve immunotherapy through liposomes (Figure 4).

#### 3.1.1. Liposome-Based Delivery of Immunostimulatory Adjuvants to DCs

The efficient activation and maturation of DCs is a prerequisite to induce proper anti-tumor immune responses. The most critical elements to achieve sufficient efficacy are immunopotentiators (enhance immune responses) and delivery systems (minimize the toxicity and enhance the efficacy). Liposomes can efficiently protect cargoes from rapid degradation and co-deliver antigens with immune adjuvants to induce powerful immune responses [56]. In addition, it has been demonstrated that liposomes are able to prolong or accelerate their release profiles when being designed and formulated accordingly [57,58]. Normally, antigens and adjuvants can be encapsulated in the hydrophilic core or hydrophobic bilayers and anchored to the surface of liposomes. There are various adjuvants currently used in cancer immunotherapy such as cytokines, CpG oligonucleotide (ODN), monophosphoryl lipid A (MPLA), and lipopolysaccharide (LPS) derivatives. Liposomes carried with these adjuvants can elevate adjuvant effects especially to immune cells, reduce systemic distribution, and minimize side effects [59].

CpG ODN has been widely reported to enhance immune responses [60]. Various liposome-based delivery systems were developed to co-encapsulate CpG ODN and antigens or other stimulatory molecules to the same DCs. Murine studies exhibited that liposomes loaded with CpG ODN and poly (I:C) enhanced immunogenicity to the co-loaded antigens and to more efficient tumor control [61]. In this work, Bayyurt et al. have shown that co-encapsulation by liposomes provided nearly 2.5 fold and 5 fold more uptake by DCs of CpG ODN and of poly (I:C), respectively, than non-encapsulated ligands. Compared to mice that received “free” CpG ODN and poly (I:C) only, the mice immunized with liposomal formulations had reduced tumor size and overall survival. In addition, liposomes have the potential to further improve the immune stimulatory properties because they can be modified with various types of biomolecules to achieve different effects [62,63]. For instance, mannose modified liposomes co-encapsulated with CpG ODN and melanoma-specific TRP2180-188 peptide specifically target DCs and show synergistic effects [64,65]. In comparison to the administration of “free” drugs, these DC-targeting liposomes enhanced anti-tumor responses and mice survival owing to an increase of effector T cells. DC-based liposomes serve as excellent carrier for immune stimulatory molecules, which play an important role in improving immunotherapeutic effect. Kwong et al. demonstrated that liposomes anchored with anti-CD40 antibodies and CpG retained bioactivity in the local tumor tissue and displayed limited toxicity after being injected intratumorally in the B16F10 murine model of melanoma [66]. 

MPLA is another type of effective immune adjuvant that can be encapsulated in liposomes, which is among the first generation of Toll-like receptors agonists approved for human application [67]. Mechanistically, MPLA can trigger TLR4 signaling and further induce the expression of proinflammatory cytokines (TNFα, IL6, IL12), chemokines, and costimulatory molecules (CD80/CD86) on DCs [68,69]. However, “free” MPLA administered intravenously in humans is highly toxic [70,71]. Notably, the incorporation of MPLA into liposomes containing saturated phospholipids can strongly reduce toxicity despite high doses, because the hydrophobic regions of other lipids and cholesterol in the liposomal bilayers closed to MPLA reduce toxicity [72]. MPLA has been incorporated into liposomes with different lipids for cancer immunotherapy. For instance, liposomes containing glycoprotein 100_280-288_ peptide and MPLA induced significantly higher uptake by DCs and CD8 T cell responses compared to the co-administration of “free” MPLA [73]. In this study, both liposomes and “free” MPLA activated DCs, but cross-presentation was only improved when MPLA was co-encapsulated into the liposomes. Interestingly, only the liposomes showed a tendency to enhance the expression of CD83 on DCs, which is a costimulatory molecule of CD86 that induced a higher production of pro-inflammatory cytokines (IL6, IL8, and IL1β). Clearly, the strategy of targeting DCs by using liposomes is able to induce more potent immune responses and enhance immune surveillance with high efficiency.

In addition to their delivery capability, liposomes themselves can act as adjuvants to stimulate immune responses intrinsically [74,75,76,77,78,79,80]. The immunological properties of liposomes have been extensively investigated to enhance both humoral and cell-mediated tumor immunity [81,82]. There are various properties including lipid composition, size, charge, and surface modification that can influence delivery efficiency and immune responses [83]. Among these parameters, the surface charge of liposomes can influence their uptake, which is mainly affected by the type and ratio of phospholipid composition [84]. Interestingly, cationic liposomes (based on positively charged lipids such as 1,2-dioleoyl-3-trimethylammonium-propane, DOTAP and dimethyldioctadecylammonium) are more potent in interacting with immune cells and lead to more immune activation than anionic and neutral liposomes [85,86]. They can activate DCs directly without adding adjuvants and are promising vehicles for cancer vaccination [87]. There are various lipids with a positive charge at physiological pH that can be used to generate cationic liposomes. For example, it has been demonstrated that DOTAP can induce DCs to secrete IL-12 and CCL2 and the subsequent promotion and activation of CD8 T cells, leading to potent CTL immune responses against cancer cells [88]. Shen et al. designed DOTAP liposomes to carry immunogenic lipoprotein rlipoE7m and phosphodiester CpG to DCs, which efficiently enhanced the immune-stimulatory effects [89]. The results showed that DOTAP was necessary for the successful delivery of rlipoE7m and phosphodiester CpG, because administration without DOTAP did not activate conventional DCs and plasmacytoid DCs as well. It was found that DOTAP liposomes were able to alter TLR signaling pathways to favor a Th1 type of immune response and enhance the presentation of antigens. Similarly, Gao et al. reported that DOTAP-based liposomes promoted much more antigen cross-presentation by DCs and mediated cross-priming to CD8 T cells than anionic liposomes [90]. It was speculated that cationic liposomes mediated the alkalization of lysosomal pH in DCs and reduced antigens’ degradation. This led to the disruption of endolysosomal membranes, cytosolic delivery of antigens, promotion of antigen cross-presentation, and cross-priming of antigens.

#### 3.1.2. Liposome-Based Immunogene Therapy

Unlike peptides or protein, nucleic acids are inherently immunogenic, and there is a lesser need to use adjuvant simultaneously. The propensity of liposomes to accumulate in reticuloendothelial organs such as the liver and spleen, where antigen-presenting cells are abundant, can be exploited for the efficient delivery of antigen-coding RNA. Once foreign RNA is introduced into cells, it can be recognized by pathogen recognition receptors (e.g., Toll-like receptors), which further incites the production of type I interferon to induce potent immune responses [91,92]. 

Fusogenic liposomes are regarded as ideal candidates for RNA delivery vehicles because delivery via fusogenic liposomes is one of the early proposed solutions to the problem of endocytic sequestration and the subsequent lysosomal degradation of RNA [93]. Synthetic molecules with fusogenic or membrane disruptive activity are normally used for the construction of liposomes to achieve membrane fusion [94]. After cellular internalization, fusogenic liposomes introduce the RNA into cytosol to achieve effective cellular immunity [95]. Stremersch et al. developed anionic fusogenic liposomes (equipped with cholesteryl hemisuccinate) and assed their siRNA delivery potential in B16F10 cancer cells and in the monocyte/DC (JAWSII) cell line [96]. These fusogenic liposomes successfully delivered cargo siRNA and resulted in a significant downregulation of the target gene expression. Normally, the endosomal/lysosomal escape of RNA is a major barrier for subsequent gene transcription in the cytoplasm. Recently, a new generation of fusogenic liposomes, which immediately fuse with the cellular plasma membrane upon contact, was developed and demonstrated in Chinese hamster ovary cells (CHO-K1) and human epidermal keratinocytes (nHEKs) [97]. It might also be interesting to extend this strategy for the cytoplasmic delivery of antigen coding RNA in DCs.

RNA lipoplexes also represent a promising delivery system for DC targeting. They can protect RNA from extracellular ribonucleases and enable the systemic delivery to lymphoid tissues to induce the selective expression of their RNA cargo in resident APCs. Salomon et al. reported RNA lipoplexes encoding CD4 T cell-recognizable neoantigens and established potent adaptive T cell responses by boosting in situ CD8 T cell immunity [98]. Whereas the currently reported lipoplexes involved various modifications to target DCs, precise DC targeting in lymphoid compartments can also be achieved without surface functionalization, solely by adjusting the negative net charge of the lipoplexes [99]. For example, Kranz et al. reported lipoplexes carrying RNA that induced strong effector and memory T cell responses and mediated a potent interferon-α-dependent rejection of progressive tumors [100]. On the other hand, positively charged lipoplexes, as typically used for gene delivery, tended to accumulated in the lungs and less in the spleen, whereas the gradual decrease of the cationic charge shifted the targeting site from the lungs toward the spleen. To balance this selectivity and transfection efficiency, negatively charged lipoplexes (lipid:RNA ratio of 1.3:2) were selected to effectively target RNA to the spleen. Furthermore, these lipoplexes also showed better immune responses than local vaccine delivery, suggesting a large therapeutical potential in to improve cancer immunotherapies.

Ternary complexes comprising of cationic liposomes, cationic polymers, DNA or RNA lipopolyplexes were studied thoroughly by Huang and collaborators [101,102]. Lipopolyplexes embedded with mRNA are efficiently internalized by DCs via the clathrin-dependent endocytosis pathway and can induce potent immune responses [103]. Lipopolyplexes differ from those assembled with DOTAP or DOTMA. They are beneficial for endosome destabilization and mRNA delivery in the cytosol due to the presence of imidazolium lipophosphoramidate and an ionizable histamine lipophosphoramidate [104]. Furthermore, the histidylated and PEGylated polylysine (cationic polymer for condensing mRNA) inside the lipopolyplexes can facilitate membrane destabilization of the endosome [105]. In preclinical studies, the DC-specific lipopolyplex carrying mRNA of MART-1 showed improved prophylactic protection against B16F10 melanoma in mice [106]. Similarly, lipopolyplexes functionalized with a glycolipid containing a tri-antenna of α-D-mannopyranoside (triMN-LPR) significantly improved the DC targeting and exerted potent effects for cancer immunotherapy in different experimental tumor models [105]. The triMN-LPR possessed an advantage of improved binding to 293T DCs than naked lipopolyplexes. Interestingly, only triMN-LPR immunizations were able to significantly induce the recruitment of inflammatory DCs to draining lymph nodes. Compared to mRNA lipoplexes (mRNA + liposomes), the mRNA lipopolyplexes contain a hybrid lipid–shell polymer that resulted in strong anti-tumor T cell immunity with less adverse effects (e.g., mild flu-like symptoms and liver toxicity and related autoimmune pathologies) [107]. The differential interaction of mRNA with innate RNA sensors due to the inherently physicochemical properties of lipopolyplexes likely alters their immunogenicity and safety profile. Thielemans et al. has reported hybrid lipopolyplexes incorporated with N1 methyl pseudouridine nucleoside modified mRNA to reduce inflammatory responses without hampering T-cell immunity [108]. In this study, immunization with lipopolyplexes displayed potent T-cell immunity and superior effects in controlling tumor growth compared to mRNA and lipoplexes. The different mode of action of lipopolyplexes enabled the generation of an equally potent vaccine with less proinflammatory effects. Taken together, lipopolyplexes are an effective delivery system for RNA for the targeted delivery to DCs and potent low-inflammatory alternatives to the mRNA lipoplexes currently investigated in both preclinical and clinical trials.

#### 3.1.3. Liposome-Based Delivery of Immunostimulatory Molecules to T Cells

Intensifying the effector phase of immune responses by the direct activation of T cells or by using genetically engineered T cells is also a promising strategy to improve immunotherapy [109]. DCs can express both positive and negative costimulatory molecules for T cell activation and are crucial to overcome negative feedback signals that can impair efficient immune responses. Hence, liposomes that can directly stimulate tumor-specific T cells and reinforce cancer immunotherapy may hold great potential for therapeutic improvements. One such approach involves the assembly of liposomes that mimic natural APCs to provide persistent and strong activation and positive costimulation signals to T cells. When these artificial APCs were administered in tumor-bearing mice treated with adoptive cell transfer, potent T cell activation and proliferation in vitro and anti-tumor potential in vivo were observed [110]. Cheung et al. described a system (APC-ms) that consisted of liposomes and mesoporous silica micro-rods to achieve efficient activation of T cells [111]. The APC-ms contained peptide-loaded MHC, CD3 mAbs (for polyclonal expansion), and CD28 mAbs (for T cell activation), and the micro-rods enabled the sustained release of IL-2. Compared to conventional expansion systems (e.g., beads), these mimetic scaffolds promoted tenfold greater antigen-specific expansion of primary mouse and human T cells. Likewise, Zappasodi et al. developed another artificial APCs type of liposome composed of CD3 mAbs, CD28 mAbs, and LFA-1 mAbs (for adhesion) [112]. Experimental data in vivo showed that these liposomes expanded both polyclonal T cells and MART-1-specific CD8 T cells in a more efficient manner than other similar systems. These observations provided proof-of-principle that liposomes can directly trigger cognate T cells, bypassing the need for a processing intermediary such as DCs.

### 3.2. Liposome-Based Delivery of Immune Checkpoint Blockade Molecules for Enhancing Cancer Immunotherapy

In the last decade, harnessing the power of the immune system against cancer has become an increasingly effective therapeutic option that can result in potent and durable responses in multiple cancer types. However, the activated tumor specific T cells do not often correlate with tumor regression in patients, which goes in accordance with the discovery and characterization of immune escape mechanisms [113,114]. Immune escape is a major challenge for cancer immunotherapy, and current clinical efforts are made to overcome immunosuppressive networks and to normalize the TME suppressive state, which is driven by suppressive mediators including cytokines and specific cell population types [115,116,117]. Typical immune checkpoint blockade through liposomes focuses on switching off specific negative feedback pathways to increase immune responses. This is because the systemic administration of immune checkpoint blockade antibodies is often accompanied by serious toxicity that limits their dose and thereby efficacy [118,119]. This finding is clinically manifested with autoimmune-like/inflammatory side effects, which cause collateral damage to normal organs and tissues [120]. Liposomes have been widely used to overcome these side effects (fure 4). There are various immune-modulating receptor–ligand interactions between immune cells and cancer cells investigated as monotherapies or combinational therapies [121,122,123]. In this regard, CTLA-4, PD-1, and PD-L1 are the most well-known inhibitory immune checkpoint receptors that have paved the way for this type of cancer immunotherapy.

#### 3.2.1. Blockade of CTLA-4 Via Liposomes

During early T cell activation, CTLA-4 is expressed and then competitively inhibits CD28 binding to CD80 and CD86 on DCs, thus leading to decreased T cell survival and expansion [124,125,126]. A fully humanized anti-CTLA-4 monoclonal antibody (ipilimumab) was approved by the Food and Drug Administration in 2011 and it showed great clinical value [127]. However, the incidence of immune-related adverse events induced by CTLA-4 blockade increased up to 70% [128,129]. To reduce the side effects, PEGylated liposomes containing CTLA-4 antibodies were prepared, and they showed a better outcome than the use of antibodies alone. In comparison to “free” antibody, liposomes showed higher accumulation in the tumor site, improved therapeutic responses, and lowered toxicity in other organs [130].

#### 3.2.2. Blockade of PD-1 Via Liposomes

The PD-1/PD-L1 pathway impairs T cell responses and induces T cell anergy, exhaustion, or apoptosis upon engagement. PD-1 is expressed on a large proportion of tumor-infiltrating lymphocytes in many different tumor types, and increased PD-1 expression is linked to tolerance to avoid auto-immunity [131,132]. PD-1 inhibits signaling downstream of the TCR and maintains peripheral tolerance by mechanisms fundamentally distinct from those of CTLA-4 [133,134]. Anti-PD-1 antibody therapy is capable of blocking the binding between PD-1 and PD-L1 and further maintains the antitumor function of CTLs [135].

To assist anti-PD-1 therapy, various effective approaches that employed liposomes have been developed. Lang et al. reported an effective liposomal system that incorporated the PD-1 inhibitor HY19991 and thioridazine into the double-layer structure of liposomes [136]. The liposomes were designed to release their cargos in the metalloproteinases-abundant region in tumors, and they increased the accumulation of intratumoral HY19991 and thioridazine by 3.65 and 7.23 fold, respectively, compared to “free” drugs. Similarly, Du et al. loaded PD-1 mAbs on the surface of liposomes and incorporated two imaging agents (IRDye800CW and ^64^Cu) and doxorubicin to simultaneously treat and track therapy progression in a breast cancer model [137]. The accumulation of PD-1 liposomes was higher than IgG control and showed better near-infrared and positron emission tomography imaging.

#### 3.2.3. Blockade of PD-L1 Via Liposomes

PD-L1, expressed on the surface of APCs and malignant cancer and tumor-associated cells, can be upregulated to a high level in various cell types by proinflammatory cytokines, particularly IFN-γ [138]. Merino et al. developed liposomes coupled with PD-L1 mAbs by two methods (conventional and post-insertion) to modulate the immune system [139]. Liposomes containing 5% PEG and prepared with the post-insertion method showed the highest cell interaction in all tested time points. It has been demonstrated that liposomes that carried PD-L1 mAbs and other molecules to Treg cells achieved potent inhibition of primary and metastatic tumors [140,141]. Furthermore, Hei et al. developed liposomes with catalase inside them and anti-PD-L1 on their surface and obtained superior therapeutic effect with low systemic toxicity [142]. Gu et al. compared “free” PD-L1 antibodies with PD-L1 liposomes and reported less tissue damage with the PD-L1 liposomes in mice melanoma model, illustrating the potential of these liposomes to reduce toxicity [143].

There are still several other immune checkpoints under exploration as potential therapeutic targets, such as the inhibitory receptors lymphocyte activation gene 3 protein [144], T cell immunoglobulin and mucin domain-containing 3 [145], and T cell immunoreceptor with Ig and immunoreceptor tyrosine-based inhibition motif domains [146]. Studies that focus on overcoming their limitations through liposomes are currently very limited, which might change soon, as it could be a promising direction to explore as possible new therapeutic options for tumors currently not responding to CTLA-4 or PD-1/L1 therapy.

### 3.3. Liposome-Based Delivery of Small Molecules to Selectively Modulate the TME

In addition to cell interaction that contributes to a suppressive microenvironment, there are some soluble mediators with complex mechanisms that can also inhibit anti-tumor immunity. Genetic alterations in tumors facilitate their growth and invasion into the surrounding tissue and also orchestrate the persistence of chronic inflammatory mediators [147]. These mediators can also modulate tumor development and progression, and they can even create an unfavorable environment for infiltrating effector cells within the tumor. Examples of some typical mediators include indoleamine 2,3-ioxygenase (IDO), transforming growth factor-β (TGF-β), adenosine, and IL-10.

#### 3.3.1. IDO

The inhibition of IDO presents a promising approach to relieve the immunosuppressive state of the TME. In the TME, IDO is generated by cancer cells, tumor-associated macrophages, myeloid-derived suppressor cells, and others. IDO catalyzes the degradation of tryptophan and production of tryptophan metabolites that limits T cell function [148]. A recent study reported by Muller et al. described that the upregulation of IDO expression can attract Tregs, which in turn inhibit anti-tumor responses significantly [149]. Additionally, in this setting, the lack of specificity and effectiveness in therapies that aim to inhibit IDO provides possibilities for liposomes. For example, murine studies demonstrated that mannose-conjugated liposomes with encapsulated IDO siRNA could efficiently and preferentially silence IDO expression in APCs [150]. Using the B16-F10 melanoma model, the authors showed that these liposomes protected T cells in the tumor from apoptosis and restricted the Treg population in both the tumor-draining lymph node and spleen. Recently, it has been reported that the liposome-mediated IDO inhibition may synergize with chemotherapeutics and immune checkpoint blockade therapy [151]. In particular, when combined with immunogenic cell death-inducing chemotherapeutics, it could provide synergistic effects in the treatment [152]. By using liposomes, these drug combinations can be simultaneously delivered to the desired sites, and thereby enhanced treatment efficacy can be achieved with low systemic toxicity. In a study, liposomes loaded with doxorubicin and indoximod (i.e., an IDO-1 pathway inhibitor) were used to conduct pharmacokinetics, efficacy, and safety studies in a murine orthotopic model that resembles human triple negative breast cancer [153]. The dual-delivery liposomes significantly augmented the antitumor immune responses against the primary as well as metastatic tumor sites.

#### 3.3.2. Adenosine

Adenosine is also an important ribonucleoside and metabolite regulating immune function that can be detected in the TME. The hypoxia in the TME can induce the upregulation of extracellular adenosine signals through the adenosine receptor A_2A_. A_2A_ is a receptor that is expressed on the surface of CD8 tumor-infiltrating T cells, myeloid-derived suppressor cells, and natural killer (NK) cells to regulate immune response during (chronic) inflammation. Emerging studies have shown that blocking the adenosine-A_2A_ pathway through pharmacologic inhibition or genetic silencing could significantly reverse the immune suppression in the TME by improving the function of T cells and NK cells in multiple tumor models [154]. SCH family is one of the most selective and potent adenosin–A_2A_ pathway antagonists, and it cannot be applied widely in the clinic due to its poor pharmacokinetic profile and hydrophobic nature [155]. Liposomes capable of maintaining high drug accumulation in the tumor site and minimizing off-target toxicity toward normal tissue are an ideal choice to overcome this pharmacokinetic barrier. Siriwon et al. designed a liposomal platform loaded with SCH via chemical conjugation to achieve better immunotherapeutic effects [156]. In this study, liposomes could attach covalently to T cells without influencing the cells’ normal function and increase the number of tumor-infiltrating lymphocytes due to the A_2A_ blockade. However, switching off adenosine–A_2A_ pathway directly can also induce undesired effects such as increased tissue inflammation and damage [155]. An alternative approach is found to downregulate adenosine by the inhibition of the ectonucleotidases CD39 or CD73, both of which are associated with the regulation of extracellular adenosine [157]. Allard et al. described an additive activity when a CD73-specific antibody was combined with CTLA-4 or PD-1 antibodies, which demonstrated strong activity because adenosine upregulated PD-1 expression on the target cells [158]. Based on such indications, it is possible to design and prepare liposomes related to adenosine to achieve better efficacy in the near future.

#### 3.3.3. TGF-β

The intervention in the TGF-β pathway provides a good opportunity to augment the anti-tumor efficacy using liposomes. In a pancreatic ductal adenocarcinoma model, Meng et al. developed PEGylated liposomes with TGF-β inhibitors and observed a decrease in pericyte coverage of the tumor vasculature, which allowed higher access of liposomes to the tumor site [159]. As a pleotropic cytokine present in TME, TGF-β is associated with multiple functions such as angiogenesis as well as immunosuppression [160]. It is produced by stromal cells, and its cleavage involves extracellular matrix adhesion and G-protein-coupled receptor-mediated integrin activation, which makes it a promising target for immunotherapy [161]. Xu et al. studied the delivery of siRNA against TGF-β using liposome–protamine–hyaluronic acid combined with vaccination (tumor antigen Trp 2 peptide) [162]. The authors showed approximately 50% reduction of TGF-β in the TME, which boosted vaccine efficacy and inhibited more tumor growth than vaccine treatment alone. Thus, liposome-based delivery systems for TGF-β modulation provide a powerful tool for local immune modulation without significantly interrupting its systemic functions.

Motivated by such findings, several small molecule inhibitors targeting TGF-β ligands or receptors have started undergoing clinical trials [163]. However, the low overall response rate and the increased risk of autoimmune diseases by systemic inhibition largely limit the application of TGF-β inhibitors [164,165]. One strategy to improve their efficacy while minimizing side effects is to selectively deliver these therapeutics to immune effector cells or to the TME. Park et al. showed that PEGylated liposomes encapsulated a small molecule TGF-β inhibitor together with IL-2 significantly delayed tumor growth due to the increased activity of NK cells and the infiltration of activated CD8 T cells in a B16/B6 mouse model of melanoma after intratumoral or systemic administration [166]. The pharmacokinetic profile displayed sustained delivery of the drugs from the localized depot of liposomes (injected peritumorally) to both tumor mass and TME gradually. Critical to the success of this combination therapy is that liposomes can sustainably release drugs with different physiochemical properties in the tumor. In addition, Zheng et al. explored the potential of targeting antitumor lymphocytes directly in vivo using liposomes encapsulated with a potent TGF-β small molecule inhibitor to adoptive T cells [167]. More sustained TGF-β inhibition could be achieved via liposomes bound to the adoptive T cells surface that continuously releases loaded drugs or via liposomes that are internalized and degraded in the endolysosomal pathway. Accordingly, liposomes show promising potential to optimize the delivery of drugs to these important immune effectors.

### 3.4. Liposome-Based Delivery of Combinational Therapy for Improving Cancer Immunotherapy

Increasing evidence indicate that immunotherapies are effective in multiple cancer types, and therefore, the combination of immunotherapy with other therapies may be beneficial in a broad range of tumors. Liposomes integrated with different therapies and loaded with multiple drugs can not only facilitate anti-tumor effects but also modulate the tumor immune environment efficiently compared to monotherapy only. The choice of therapeutic agents and timing of these combinations is critical, because different agents normally show different mechanisms and target sites. Combinatorial therapy can be a double-edged sword, as it can elicit potent anti-tumor effects and also increase the risk of systemic toxicity. In this regard, liposomes show great potential to overcome the aforementioned limitations, as they can simultaneously deliver agents with different physicochemical properties and mitigate adverse effects. A good liposome-based delivery system needs to meet the following requirements: (1) co-load different molecules in sufficient concentrations; (2) overcome biologic barriers without losing its bioactivity; (3) release cargos at the desire site and time; (4) have the ability to target specific tumor or cell type; (5) exhibit synergistic or additive effects; and (6) must utilize economic, efficient, and safe preparation methods.

#### 3.4.1. Liposome-Based Delivery of Immunochemotherapy

Immunochemotherapy might be one of the typical strategies with intriguing results that indicate that chemotherapy can enhance the therapeutic outcome of immunotherapy and further reverse chemoresistance [168,169,170]. Several studies reported that low-dose chemotherapeutics were able to increase the susceptibility of cancer cells to CTLs [171,172,173]. However, the success of immunochemotherapy is limited by the lack of an efficient platform for the efficient co-delivery of drugs to tumors. A successful liposome-based immunochemotherapeutic solution can accomplish the following aims at the same time: release cytotoxic drugs in specific sites as well as prime and enhance anti-tumor immune cell populations. For example, liposomes with matrix metalloproteinases-responsive behavior loaded with a PD-L1 inhibitor and (low dose) doxorubicin achieved better anti-tumor efficacy than the components administered separately [174]. By adjusting the mix ratio of lipids with pH responders, liposomes could be fine-tuned to optimize the response sensitivity and specificity. A low dose of doxorubicin and hydrolysis resistant D-peptide as an antagonist to target the PD-1/PD-L1 pathway were co-encapsulated in liposomes to sensitize tumor cells with negligible systemic side effects. These liposomes were relatively stable in physiological conditions and released loaded cargos in target sites once triggered by acidic pH at the tumor site, which facilitated the recruitment of tumor-infiltrating lymphocytes resulting in stronger tumor inhibition. Furthermore, Yang et al. designed TH peptide-modified liposomes that could shield antigen (αGC) from the uptake by B cells and trigger antitumor responses [175]. Paclitaxel released from liposomes could further help the release of TAAs into the surrounding environment, thereby strengthening the specific antitumor immunity of the immunotherapy in melanoma-bearing mice.

#### 3.4.2. Liposome-Mediated Immunotherapy Combined with Sonodynamic or Phototherapy

Sonodynamic therapy involves the combination of low-intensity ultrasound with sonosensitizers for high tissue-penetrating capability to generate reactive oxygen species to induce cell death and subsequent immune responses against the released TAAs [176,177]. Yue et al. incorporated the sonosensitizer hematoporphyrin monomethyl ether and the immune adjuvant imiquimod into liposomes as a promising sonodynamic therapy-based immunotherapy [178]. This approach could delay the growth of primary tumors and inhibit both primary and distant tumor once combined with PD-L1 blockade in the CT26 colorectal cancer and the 4T1 breast cancer models. 

Phototherapy (i.e., photodynamic therapy and photothermal therapy) can also be enhanced with rationally designed liposomes for the targeted delivery of photosensitizers and by a process similar to that of sonodynamic therapy. It produces reactive oxygen species to destroy cancer cells and stimulate the immune system through the release of TAAs [179,180,181]. Since traditional photosensitizers are normally untargeted and poorly soluble, a liposome-based combination of phototherapy and immunotherapy can improve the therapeutic index of these modalities by enhancing the stability and biocompatibility of cargos as well as reducing sides effects [182,183].

Shi et al. devised a liposome-based immune stimulation strategy to immune regulate the TME and to significantly augment tumor growth inhibition when combined with photodynamic therapy [184]. Kim et al. have shown tumor-targeting efficiency, and the prevention of drugs leakage from liposomes can be improved by functionalizing the liposomes with the photosensitizer KillerRed-embedded cancer cell membrane combined with photodynamic therapy. This effectively induced potent anticancer immune responses, inhibited the growth of the primary tumors, and reduced the number of lung metastasis in tumor-bearing mice [185]. Unlike conventional liposomal carriers used for cancer therapy, these liposomes did not exhibit unwanted drug leakage and had an outstanding cancer-targeting efficiency. Wang et al. reported a liposome delivery system containing the lipophilic photosensitizer Ce6, and low molecular citrus pectin enhanced photodynamic therapy, which resulted in NK cell-related immune activation in melanoma [186]. After administration, the photodynamic therapy-induced tumor cell apoptosis was improved by the Ce6 photosensitizer encapsulated in the liposomes, whereas the invasion and metastasis of cancer cells after photodynamic therapy were controlled by the citrus pectin. Furthermore, incorporation of the MPLA immune adjuvant in the liposomes strengthened the anti-tumor effect following photodynamic therapy.

Recently, several studies have reported that the combination of nanocarriers with photothermal therapy and immunotherapy could generate higher antitumor immunological effects than without nanocarriers [187,188,189]. To achieve this efficacy, one of the key points is to realize consistent tumor accumulation, which can be greatly improved by the utilization of liposomes. Li et al. designed an IR-7-loaded liposome coated with CpG oligonucleotides for photothermal therapy-mediated immunotherapy in the CT26 colon cancer model [190]. After the combinational treatment, the percentage of myeloid-derived suppressor cells and Treg cells in the tumor remarkably decreased, which implied that the combinational therapy decreased immunosuppression in the TME and that liposomes were essential for this effect.

## 4. Conclusions

Many great achievements of liposome-based drug delivery systems for cancer immunotherapy in recent years have been reported, and there is little doubt that liposomes provide a promising platform to improve cancer treatments via different immunotherapeutic mechanisms. However, further optimization of liposomes is required tuned to their specific purposes. It is crucial to design liposomes to precisely target the specific sites in the TME and to reduce off-target effects, which lead to poor outcomes that can be more unpredictable than traditional therapies. Areas for further improvement are (1) improving the pharmacokinetics of liposomes to reduce biodistribution to achieve the least possible toxicity, and (2) adjusting the liposomes according to the purpose of the specific immunomodulator and its target. In summary, liposomes exhibit distinctive advantages for cancer immunotherapy, such as high safety, efficient delivery for multiple drugs, as well as inducing immune activation by themselves; therefore, they are likely to play an important role in future immunotherapeutic cancer strategies.

## Figures and Tables

**Figure 1 pharmaceutics-12-01054-f001:**
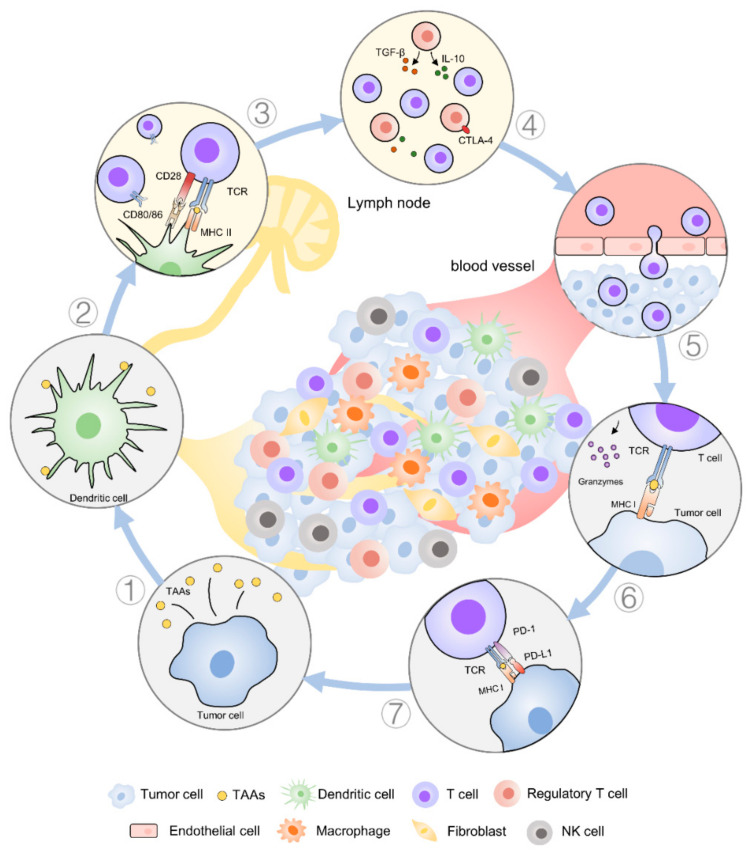
Overview of a typical build-up of an immune response against cancer cells in solid tumors. (1) Cancer cells can release tumor-associated antigens (TAAs) in the tumor microenvironment (TME). (2) Anti-tumor response is initiated with the recognition of TAAs presented by antigen-presenting cells (APCs) such as dendritic cells (DCs). (3) The cognate T cell receptor (TCR) binds to major histocompatibility complex (MHC) I/II receptor containing the epitope peptide from TAAs. The priming of T cells generally occurs in lymphoid tissue. (4) During priming, T cells are susceptible to immune negative/positive factors that prevent/promote their full activation mediated by cytokines (e.g., transforming growth factor-β (TGF-β) and interleukin 10 (IL-10)) and costimulatory receptors, such as cytotoxic T-lymphocyte-associated protein 4 (CTLA-4). (5) Once activated successfully, effector T cells proliferate, secrete inflammatory cytokines, acquire cytolytic properties, and migrate to tumor sites. (6) Cytotoxic T cells can identify cancer cells and bind to cognate cancer antigens presented on MHC I on cancer cells and initiate T cell-mediated killing (e.g., release granzymes). (7) T cell function can also be stimulated or inhibited in the tumor. Negative costimulatory signals (e.g., programmed cell death protein 1 ligand 1, PD-L1) inhibit the function of T cells and induce anergy and the exhaustion of T cells.

**Figure 2 pharmaceutics-12-01054-f002:**
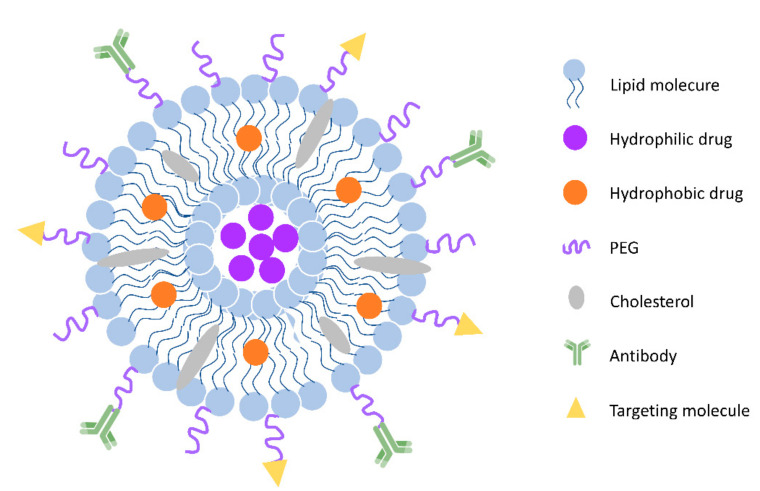
General scheme of liposomes. Liposomes are spherical vesicles with a hydrophilic core formed by a phospholipid and cholesterol bilayer. They can also be modified with polyethylene glycol (PEG) coating for long circulation and various molecules (peptides, antibodies, et al.) for targeting.

**Figure 3 pharmaceutics-12-01054-f003:**
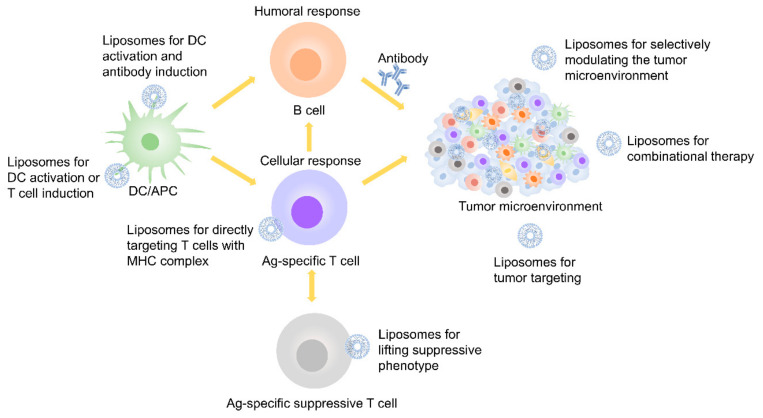
Liposome-based treatment used in immunotherapy.

**Figure 4 pharmaceutics-12-01054-f004:**
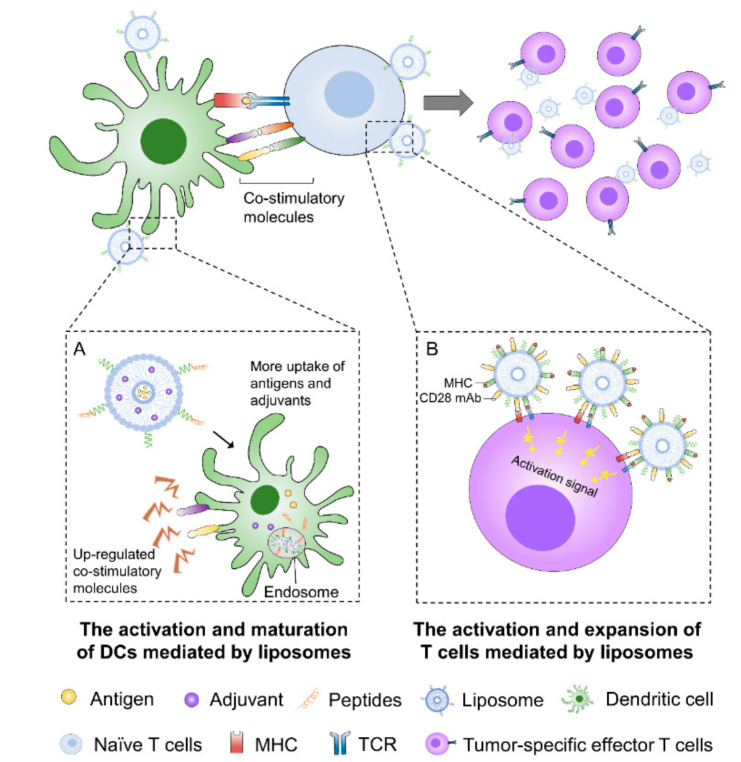
Liposome-based delivery systems of immunostimulatory molecules to DCs and effector T cells. (**A**) Liposomes can target DCs directly by receptor-mediated internalization. The induction of immune responses mediated by liposomes can increase the efficacy of cytotoxic T lymphocytes (CTLs) due to an enhanced uptake of antigens and adjuvants by DCs resulting in a higher upregulation of costimulatory receptors and the maturation of DCs. (**B**) Liposomes can activate cytotoxic CTLs and intensify the effector phase of immune responses. Direct T cell triggering can be achieved by liposomes that mimic natural APCs to provide persistent and strong activation as well as positive costimulation signals to T cells.

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
