# Peer review of "Liposome-Based Drug Delivery Systems in Cancer Immunotherapy"

_pharmaceutics, 2020, doi:10.3390/pharmaceutics12111054_

Round 1

Reviewer 1 Report

In this work, Gu and co-workers described lipid-based systems available for cancer immunotherapy. The work is well organized and described all the possible appraches in the modulation of immune system in cancer therapy. A general check for typos and grammar error should be performed on the entire manuscript. I have one major concern regarding a review that I found (doi: 10.1038/s41401-019-0281-1): can the authors state that these two reviews are actually different? Minor concern: Please check that every statement throughout the work is supported by the right references.

Author Response

We thank the reviewer, for his/her kind words and bring this review to our attention.

We have gone thoroughly through our whole manuscript and corrected typos and grammatical errors. Regarding the proposed review, although there is some minor overlap (both reviews comprise the themes liposome technology and immunotherapy), in our opinion that review article is substantially different from ours.

In our manuscript we specifically cover different areas and aspects that are not described in the mentioned review. For instance, liposome-based delivery of small molecules to selectively modulate the tumor microenvironment. And we cited it in the manuscript, see line 143. Furthermore, the indicated review is especially focused on the liposomal effects on the molecular aspects of individual immune cells, whereas our manuscript is more focused on the broader aspects of liposomal nanotechnology and their applications to which problems they may provide solutions, including for the immunotherapy of cancer.

Reviewer 2 Report

Reviewer's comments:

Remarks to the Author (Pharmaceutics-968955-peer-review-v1)

It is not a good idea to write a review paper without enough information and the authors did not completely cover the areas designated in the review article.

Problems with this manuscript.

  • The review article is poorly written. A simple collection of published papers and listing them in the figures do not help readers at all and these figures are not well organized, either: The content of the table is a collection of literature.
  • The quality of all the figures was very poor.

Conclusion and perspectives: I suggest authors rewrite this section; including future investigations needed in liposome-based micro or nanosystems for their clinical translation. 

Besides, there are review papers on liposome-based drug delivery systems for cancer and other biomedical applications. For instance,

  1. Materials Science and Engineering C 71 (2017) 1327–1341; DOI: 10.1016/j.msec.2016.11.073

The above article has already explored the liposome-based drug delivery systems in cancer, which is very similar to the current review draft. Surprisingly, the authors did not cite this paper in the current review draft. The authors need to describe prior published review papers in the introduction; about how their review article is different, and what additional information it is providing other than previously published review articles on liposomes.

The review is a chance to give a perspective and integrate different information published more compared to the only description of results from different authors. I do not see the purpose of the review and there is a lack of a proper discussion about the liposome as a drug delivery system according to the releasing/delivery target and drug used.

Author Response

We thank this reviewer for carefully reading and commenting on our review manuscript, which we have used to improve our manuscript. Besides, we are happy with the proposed new literature, which we have discussed and added to the bibliography of our manuscript. However, we regret that this reviewer does not see the added value of our review, as opposed to reviewer 1 and 3.

We understand that there are other review publications with overlapping subjects, but our manuscript is substantially different from other published articles. For example, the review proposed above focuses primarily on the delivery of chemotherapeutic drugs by liposomes, whereas our manuscript is focused on the broader aspects of liposomal nanotechnology and their applications for different forms of immunotherapy of cancer.

Reviewer 3 Report

Manuscript ID: pharmaceutics-968955

The manuscript “Liposome-based Drug Delivery Systems in Cancer Immunotherapy” prepared by Zili Gu, C.G. Da Silva, Koen van der Maaden, Ferry Ossendorp and Luis J. Cruz as Co-authors reviewed different liposomal systems developed for immunomodulation in cancer. The manuscript is well written; it covers an important area and is informative and useful for other scientists.

However, there are some issues needed to be addressed before publication.

  1. The abstract contains a large amount of basic information which is not related directly to the Please re-write it, and put more stress on the main highlights regarding your review.
  2. Please re-write title of subparagraph 1 and don’t use verb in titles.
  3. Please restructure text and shift Figure 1 nearly to place when it was mentioned for the first time.
  4. Lines 151-153 authors wrote: “The inherently physicochemical properties of liposomes including their size, charge, polarity and configuration may also have negative impacts on the enhanced permeability and retention (EPR) effect and related processes [43,44]” Please justify what relationships between properties of liposomes and their impact on processes.
  5. Please improve quality for Figure 3.
  6. Please use the same style for the description of term poly(I:C).
  7. In conclusions authors wrote: “Based on the reported achievements of liposome-based drug delivery systems for cancer immunotherapy in the last 5 years, ...” However from 170 literature sources only 68 ones represent the time from 2015.
  8. Please to re-write conclusions and give more detailed perspectives instead of general information.

Consequently, I do recommend accepting this manuscript for publication with minor revision.

Author Response

We thank this reviewer for his/her extensive comments on our manuscript, which we have used to substantially improve our manuscript. Furthermore, we are grateful for the acknowledgment that our manuscript is informative, covers important areas, and is useful for other scientists, which was exactly our aim: to report the status of current progression, to illustrate the benefits and highlight possible pitfalls of using liposome nanotechnology for the immunotherapy of cancer.

  1. The abstract contains a large amount of basic information which is not related directly to the Please re-write it, and put more stress on the main highlights regarding your review.

We thank the reviewer for pointing this out. We have amended the abstract to be more to the point and emphasized the highlights of the manuscript, see line 12-28.

  1. Please re-write title of subparagraph 1 and don’t use verb in titles.

We have amended the title. See line 36/37.

  1. Please restructure text and shift Figure 1 nearly to place when it was mentioned for the first time.

We have shifted Figure 1 to the nearest place where it was mentioned for the first time, see line 53-66.

  1. Lines 151-153 authors wrote: “The inherently physicochemical properties of liposomes including their size, charge, polarity and configuration may also have negative impacts on the enhanced permeability and retention (EPR) effect and related processes [43,44]” Please justify what relationships between properties of liposomes and their impact on processes.

We have amended this sentence. See line 181-186.

  1. Please improve quality for Figure 3.

Figure 3 has been revised.

  1. Please use the same style for the description of term poly(I:C).

We have adapted the text in such a way that poly(I:C) is written down consistently, see line 257.

  1. In conclusions authors wrote: “Based on the reported achievements of liposome-based drug delivery systems for cancer immunotherapy in the last 5 years, ...” However from 170 literature sources only 68 ones represent the time from 2015.

We agree with the reviewer and have amended this sentence accordingly, see line 85.

  1. Please to re-write conclusions and give more detailed perspectives instead of general information.

The conclusion has been re-written with more included details, see line 627-641.

Round 2

Reviewer 2 Report

The manuscript is improved after the revision, and it can be publishable in Pharmaceutics.